# Optimizing the Seeding Density of Human Mononuclear Cells to Improve the Purity of Highly Proliferative Mesenchymal Stem Cells

**DOI:** 10.3390/bioengineering10010102

**Published:** 2023-01-11

**Authors:** Hiroyuki Nagai, Akihiro Miwa, Kenji Yoneda, Koichi Fujisawa, Taro Takami

**Affiliations:** 1Shibuya Corporation, Kanazawa 920-8681, Ishikawa, Japan; 2Department of Clinical Laboratory Science, Faculty of Health Science, Yamaguchi University Graduate School of Medicine, Ube 755-8505, Yamaguchi, Japan; 3Department of Gastroenterology and Hepatology, Yamaguchi University School of Medicine, Ube 755-8505, Yamaguchi, Japan; 4Department of Environmental Oncology, Institute of Industrial Ecological Sciences, University of Occupational and Environmental Health, Kitakyushu 807-8555, Fukuoka, Japan

**Keywords:** MSC, cell isolation, mononuclear cell, colony

## Abstract

Mesenchymal stem cells (MSCs) hold considerable promise for regenerative medicine. Optimization of the seeding density of mononuclear cells (MNCs) improves the proliferative and differentiation potential of isolated MSCs. However, the underlying mechanism is unclear. We cultured human bone marrow MNCs at various seeding densities (4.0 × 10^4^, 1.25 × 10^5^, 2.5 × 10^5^, 6.0 × 10^5^, 1.25 × 10^6^ cells/cm^2^) and examined MSC colony formation. At lower seeding densities (4.0 × 10^4^, 1.25 × 10^5^ cells/cm^2^), colonies varied in diameter and density, from dense to sparse. In these colonies, the proportion of highly proliferative MSCs increased over time. In contrast, lower proliferative MSCs enlarged more rapidly. Senescent cells were removed using a short detachment treatment. We found that these mechanisms increase the purity of highly proliferative MSCs. Thereafter, we compared MSCs isolated under optimized conditions with a higher density (1.25 × 10^6^ cells/cm^2^). MSCs under optimized conditions exhibited significantly higher proliferative and differentiation potential into adipocytes and chondrocytes, except for osteocytes. We propose the following conditions to improve MSC quality: (1) optimizing MNC seeding density to form single-cell colonies; (2) adjusting incubation times to increase highly proliferative MSCs; and (3) establishing a detachment processing time that excludes senescent cells.

## 1. Introduction

Mesenchymal stem cells (MSCs) are an attractive cell source for regenerative medicine because they can be harvested from various tissues, such as bone marrow, adipose, and umbilical cord. They have been proven to be clinically successful in regenerative medicine [1,2]. Owing to their ability to differentiate into cells of various lineages such as osteoblasts, adipocytes, and chondrocytes, they are extensively used in tissue engineering [3,4]. Large-scale culture techniques and efficient differentiation methods have been developed for the use of MSCs in the repair and treatment of defects, such as those in the bone and cartilage. Furthermore, due to the paracrine effect, healing has been reported in diseases, such as cerebral and myocardial infarction [5,6]. MSCs also possess anti-inflammatory and immunomodulatory properties. In recent years, extracellular vesicles produced by MSCs have been attracting attention, and research into the practical application of cell-free therapeutics is in progress [7]. These characteristics have led to their rapid and increasingly critical use in clinical practice. In our laboratory, we reported autologous bone marrow cell infusion (ABMi) therapy for liver cirrhosis [8], and we are currently developing a minimally invasive liver regenerative therapy using bone-marrow-derived MSCs (BM-MSCs) [9,10,11,12]. This therapy is currently being evaluated in investigator-initiated clinical trials (UMIN-CTR Trial No. UMIN000041461).

The use of MSCs has resulted in an increase in the number of manufacturing facilities. However, there are several critical points and concerns regarding production [2]. For example, the proliferative and differentiation potential of MSCs vary considerably depending on the age and health status of the donor. Increasing donor age decreases proliferative capacity, differentiation ability, and immunosuppressive capacity [13]. As aging progresses, oxidative stress, DNA damage, and epigenetic changes such as DNA methylation and histone modifications are accumulated [14,15,16]. These changes affect the expression of genes involved in cell cycle arrests, such as *p16* and *p21*, and the regulation of differentiation, such as Wnt signaling and *Runx2*. As a result, cellular proliferative and differentiation potential decrease. This could result in prolonged manufacturing periods or even production failure. Additionally, known variations in cell quality adversely affect therapeutic efficacy. To solve these problems, a method that stabilizes production must be established.

To accomplish this, we focused on the isolation of MSCs. Previous reports state that MSC yield can be increased by optimizing the seeding density of mononuclear cells (MNCs) [17,18,19]. However, the mechanism by which this effect occurs has not yet been elucidated, and no explanation exists regarding its relationship with cell quality. This makes it difficult to determine the important parameters of the MSC manufacturing process.

In this study, we found a new way to improve the purity of MSCs with excellent proliferative and differentiation potential by optimizing MNC seeding density. We aimed to investigate the relationship between seeding density and MSC colony formation, analyze the characteristics of MSCs in each colony, isolate MSCs at higher and lower seeding densities, and compare their proliferative abilities and differentiation potential. Our research findings provide new insights into the MSC isolation process, which will lead to improved MSC manufacturing efficiency and quality.

## 2. Materials and Methods

### 2.1. Materials

Human BM-MNCs were purchased from Lonza (Basel, Switzerland). The MNCs were extracted from the whole bone marrow of healthy donors via Ficoll density gradient separation (GE Healthcare, ORD, USA), which was then frozen. In our study, we used five lots. Donors included HIV/HBV/HCV-negative women and men with an age range of 22–35.

### 2.2. Isolation and Culture of Human BM-MSCs

MNCs were thawed and mixed with Dulbecco’s Modified Eagle’s Medium (DMEM) (Thermo Fisher Scientific, Waltham, MA, USA) supplemented with 10% fetal bovine serum (FBS) (Nichirei Bioscience, Tokyo, Japan) and 5 µg/mL gentamicin (Takada Pharmaceutical, Saitama, Japan). Cells were seeded in culture vessels at various densities and cultured in 5% CO_2_ at 37 °C (CO_2_ incubator, PHC, Tokyo, Japan). The medium was changed every 2–3 d and replaced with Stem Fit for MSC medium (Ajinomoto, Tokyo, Japan) supplemented with 0.2 µg/mL iMatrix-511 (Nippi, Tokyo, Japan) on day 7 of culture. When MSC colonies reached a high density, they were washed with Dulbecco’s phosphate buffered saline (Thermo Fisher Scientific) and trypsin (TrypLE Select Enzyme (1×), Thermo Fisher Scientific). Thereafter, cells were detached for 5 min at 37 °C. Detached cells were centrifuged for 5 min at 300× *g* (AX series centrifuge, TOMY, Tokyo, Japan). Following removal of the supernatant, the pellet was resuspended in fresh medium and seeded into new vessels at 5000 cells/cm^2^.

### 2.3. Time-Lapse Imaging

MNCs cultured on culture plates were observed every 24 h by multi-area time-lapse imaging microscopy (EVOS, Thermo Fisher Scientific). From time-lapse images, we classified “the fast-growing cells” and “the slow-growing cells” based on the timing at which the MSC reached high density, or doubling time. Doubling time was calculated based on the time taken and the number of cell counts. We counted the number of MSC by using ImageJ software (NIH).

### 2.4. Senescence-Associated β-Galactosidase Staining

Senescence-associated β-galactosidase (SA-β-gal) staining was used to detect senescent cells. Staining was performed using a commercial kit (Senescence Detection Kit, BioVision, Boston, MA, USA). Briefly, cells were fixed in formalin. Thereafter, the substrate X-gal and the staining solution were added. Following incubation for 24 h at 37 °C, cells showing SA β-gal activity turned blue. We microscopically observed the stained cells (EVOS).

### 2.5. Flow Cytometry

Cells frozen at passage 1 and 2 were thawed, blocked with FcR Blocking Reagent (Miltenyl Biotec, Bergisch Gladbach, Germany) at 4 °C for 15 min, and labeled with a fluorescent-dyed antibody. Dead cells were stained with 7-AAD (BD Biosciences, Franklin Lakes, NJ, USA) and analyzed using a flow cytometer (FACS Verse; BD Biosciences). The anti-human antibodies (CD45, CD73, CD90, and CD105) were purchased from BioLegend (San Diego, CA, USA).

### 2.6. Single-Cell Cloning Assay

MNCs were thawed, mixed with DMEM supplemented with 10% FBS and 5 µg/mL gentamicin, and seeded in 96-well plates (Corning, NY, USA) at a density of 1 × 10^4^ cells/well. As MSCs were present in 0.001–0.01% of BM-MNCs [20], we assumed that seeding at this density would result in the formation of one colony/well. The same procedure described above was used for the isolation and culture of BM-MSCs.

### 2.7. Colony-Forming Assay

Cells at passages 2 and 3 were seeded at 100 cells in a 60 mm dish (Corning) in DMEM supplemented with 10% FBS and 5 µg/mL gentamicin. After 14 d of culture, colonies were stained with crystal violet and the number of colonies was manually counted.

### 2.8. Quantitative Real-Time PCR

Total RNA was extracted from cells using a commercial kit (KURABO, Osaka, Japan). It was reverse-transcribed to cDNA using a commercial kit (TOYOBO, Osaka, Japan). Quantitative PCR was performed using FastSYBR™ Green Master Mix (Applied Biosystems, MA, USA) and 1 mL aliquots of the RT reaction. Amplification of samples was carried out in a final volume of 20 µL using the StepOnePlusTM system (Applied Biosystems) with the following program: initial denaturation at 95 °C for 20 s, followed by 40 cycles of annealing at 60 °C for 30 s, and denaturation at 95 °C for 3 s. The following primers were used (Table 1): *Ki67*, *PCNA*, and *β-actin*. To calculate relative fold-change values, CT values were standardized using *β-actin* as an internal control.

### 2.9. Serial Analysis of Gene Expression (SAGE)

For transcriptome analysis, RNA was prepared from the MSC obtained under low MNC density (1.25 × 10^5^ cells/cm^2^) and high MNC density (1.25 × 10^6^ cells/cm^2^). The total RNA was isolated from cells using Maxwell RSC simplyRNA Cell Kit (Promega, Madison, WI, USA). Ion Ampliseq Transcriptome Human Gene Expression Kit (Thermo Fisher Scientific, Waltham, MA, USA) was used for library creation. An Ion Proton next-generation sequencer library of analysis beads was created, and an Ion 540 Kit (Thermo Fisher Scientific, Waltham, MA, USA) was used for sequencing using an Ion Proton next-generation sequencer. Gene set enrichment analysis (GSEA) were integrated by R programming software. Protein-coding differentially expressed genes (DEGs) in MSCs obtained by lower MNC density (1.25 × 10^5^ cells/cm^2^) were identified based on a fold change cutoff of 2.0 in comparison with MSCs obtained by high MNC density (1.25 × 10^6^ cells/cm^2^).

### 2.10. Differentiation Assay

For adipogenic differentiation, cells at passage 3 were seeded at 2.1 × 10^4^ cells/cm^2^ in 24-well plates (Corning) according to the manufacturer’s instructions (R&D Systems, Minneapolis, MN, USA). Cells were cultured in differentiation medium for 21 d. The medium used was MEM α (Thermo Fisher Scientific) supplemented with 10% FBS, 1 × addipogenic supplement (hydrocortisone, isobutylmethylxanthine, indomethacin), 100 U/mL penicillin, and 100 µg/mL streptomycin. Thereafter, they were fixed and stained with fatty acid-binding protein 4 (FABP4) antibody (R&D Systems) and Hoechst 33342 (Thermo Fisher Scientific, Waltham, MA, USA). For osteogenic differentiation, cells at passage 2 were seeded at 4.2 × 10^3^ cells/cm^2^ in 48-well plates (Corning) and cultured in a commercial differentiation medium (Sartorius, Göttingen, Germany) for 21 d. Cells were stained with Alizarin Red and Hoechst 33342. Alizarin Red staining was performed using a commercial kit (COSMO BIO, Tokyo, Japan). For chondrogenic differentiation, cells were seeded at 6.3 × 10^4^ and 1.25 × 10^5^ cells each in 15 mL filter tubes (Greiner Bio-One, Kremsmünster, Austria) in a differentiation medium. Spheres were formed and cultured for 21 d. The medium used was D-MEM/F12 medium (Thermo Fisher Scientific, Waltham, MA, USA) supplemented with 10 ng/mL TGF-β1 (Merck, Darmstadt, Germany), 100 nM dexamethasone (Merck), 50 µg/mL ascorbic acid (Tokyo Kasei, Tokyo, Japan), 100 µg/mL sodium pyruvate (Thermo Fisher Scientific, Waltham, MA, USA), and 1% ITS+ Premix (Corning). After culturing, formalin-fixed paraffin sections were prepared and stained with Alcian blue. Each specimen was microscopically observed (EVOS).

### 2.11. Statistical Analysis

Data are presented as the mean ± standard error. Statistical analysis was performed using JMP Pro software (SAS Institute, Cary, NC, USA). Data were analyzed using Student’s *t*-test. Differences were considered statistically significant at *p* < 0.05.

## 3. Results

### 3.1. Different Seeding Densities of MNCs Cause Differences in the Timing of MSC Colony Formation

A review of the relevant literature revealed that a wide range of MNC seeding densities (4 × 10^3^–1 × 10^6^ cells/cm^2^) have been investigated [17,18,19]. Based on these studies and manufacturing costs, we set several seeding conditions (4.0 × 10^4^, 1.25 × 10^5^, 2.5 × 10^5^, 6.3 × 10^5^, and 1.25 × 10^6^ cells/cm^2^) and cultured the cells. First, we stained the colonies with crystal violet to compare the timing of colony formation (Figure 1A). The timing tended to be earlier with higher seeding density: colonies appeared at 6.3 × 10^5^ cells/cm^2^ and 1.25 × 10^6^ cells/cm^2^ on day 7, 2.5 × 10^5^ cells/cm^2^ on day 9, 1.25 × 10^5^ cells/cm^2^ on day 11, and 4.0 × 10^4^ cells/cm^2^ on day 13. Furthermore, as culture time was exceeded under each condition, cell aggregation progressed. Some colonies sloughed off from the culture vessel when cells within colonies became over-confluent. This indicated the need to collect cells at an appropriate density. As shown in Figure 1A, densification occurred faster with higher seeding densities. Thus, we expected that colonies would tend to become dense more easily when the distance between MSCs attached to the vessel is smaller. We seeded the same number of MNCs into culture vessels of different sizes and compared the colony distribution at different densities (Figure 1B). The distance between colonies increased as seeding density decreased. At a higher density of 1.25 × 10^6^ cells/cm^2^, colonies were indistinct, and the area occupied by the cells that reached high density was large. At the mid-range density of 2.5 × 10^5^ cells/cm^2^, the colonies were more than 2 mm in diameter and partially overlapped with each other. At a lower density of 4.0 × 10^4^ cells/cm^2^, colonies ranged from 1.0–6.5 mm in diameter, and cell density varied from dense to sparse (Figure 1C).

These results suggest that MNC seeding density has a notable effect on the formation of MSC colonies. Higher densities such as 6.3 × 10^5^ and 1.25 × 10^6^ cells/cm^2^ result in a reduced distance among MSCs attached to the vessel. The colonies subsequently form and overlap with each other, limiting the space available for proliferation. Thus, they reached the maximum cell density more quickly. Alternatively, where lower density conditions of 4.0 × 10^4^ and 1.25 × 10^5^ cells/cm^2^ are utilized, colonies derived from a single cell were formed. These conditions provided sufficient space for proliferation. Therefore, the densification inside these colonies was delayed.

### 3.2. At Lower MNC Seeding Density, the Superiority of Proliferative Ability Results in Significant Differences in Numbers between Cells of Different Quality

Colonies formed under low seeding density conditions showed individual differences in size and density (Figure 1C). We focused on diversity and examined the proliferation of MSCs within individual colonies over time using microscopy. Under the low seeding density condition (1.25 × 10^5^ cells/cm^2^), there was a notable variation in the number of cells between positions. The differences between positions became more pronounced as time progressed (Figure 2A,C). We calculated the doubling time and ratio of cells in each position to the total number of cells (Figure 2D). From day 5 to day 9 of culture, the percentage of the fast-growing group increased, whereas the percentage of the slow-growing group decreased. This indicated that cells with a high proliferation rate were predominant in the population as time progressed. In contrast, under the high density condition (1.25 × 10^6^ cells/cm^2^), growth rate of cells per position showed a similar pattern (Figure 2B,C).

Furthermore, at a low density of 1.25 × 10^5^ cells/cm^2^, cells with low proliferation tended to enlarge with time (Figure 2A). The enlarged cells were SA β-Gal-positive, confirming that they were senescent (Figure 2E). Trypsin dissociation treatment of the senescent cells tended to take longer than 5 min. Treatment for 5 min allowed the recovery of healthy and not senescent cells (Figure 2F).

These results indicated that under low seeding density conditions, the ratio of fast-growing cells became predominant as time progressed, depending on the superiority of their proliferative ability. Additionally, our study confirmed that by adjusting the time of the dissociation reaction, senescent cells remained in the vessel. Thus, cells with superior proliferative abilities were harvested.

### 3.3. Different Seeding Densities of MNCs Affect the Number of Remaining CD45-Positive Cells

In additional colony formation, other differences in the number of remaining hematopoietic cells were observed at different seeding densities (Figure 3A). At identical numbers of medium changes, more hemocytes were present under the high density condition (1.25 × 10^6^ cells/cm^2^). In contrast, those under the low density condition (1.25 × 10^5^ cells/cm^2^) were greatly reduced. Upon collection, the percentage of CD45-positive cells was analyzed using fluorescence-activated cell sorting (FACS). The percentage was significantly reduced in the low density condition compared to that in the high density condition (*p* < 0.0001, Figure 3B).

Lower seeding density reduced the residual number of hematopoietic cells, leading to improved purity of MSCs in harvested cells.

### 3.4. Groups of Cells That Form Colonies Quickly Show High Proliferative Ability

In studies in which lower seeding densities of MNCs were evaluated, the percentage of cells with a high proliferation rate became predominant over time. We examined whether these fast-growing cells maintained their proliferative abilities after passaging. This was accomplished by using a single-cell cloning assay to evaluate the growth rate, morphology, and gene expression of each colony (Figure 4A). We observed considerable differences in the growth rate and cell morphology after passaging, which correlated with the timing of colony formation (Figure 4B–D). The groups of cells that formed colonies quickly maintained a high rate of proliferation, resulting in a uniform population with a small cell size. In contrast, the slow colony-formation group proliferated more slowly and exhibited a larger cell size and greater variability. Gene expression was evaluated using real-time PCR. The groups with faster colony formation showed higher expression of *Ki67* and *PCNA* (Figure 4E).

### 3.5. MSCs Isolated under the Low MNC Seeding Density Condition Have Higher Proliferative Ability and Improved Differentiation Potential

The low MNC seeding density condition (4.0 × 10^4^–1.25 × 10^5^ cells/cm^2^) was found to increase the purity of MSCs with high proliferative potential, and 5 min trypsin treatment was found to exclude senescent cells (Figure 2D,F). To evaluate this method on the quality of isolated MSCs, we isolated the MSCs at low MNC density (1.25 × 10^5^ cells/cm^2^) and high MNC density (1.25 × 10^6^ cells/cm^2^), which had a clear difference on colony formation. We confirmed that isolated cells in both conditions expressed an MSC-specific marker. In the low MNC density condition, the MSC purity was higher than that in the high MNC density condition (Appendix A). Next, we compared the growth rate, morphology, and gene expression of the cells obtained under each condition. The cells obtained from the low density condition showed a significantly shorter doubling time than those obtained from the high density condition (*p* < 0.01, Figure 5A,B). Furthermore, these cells exhibited a uniform population with a small cell size, similar to the faster colony-formation group observed in the single-cell cloning assay (Figure 5C,D). We observed high colony-forming ability in cells obtained under low density condition (Figure 5E,F). They also showed higher expression of *Ki67* and *PCNA*, indicating enhanced proliferative activity (Figure 5G). SAGE analysis was performed to analyze the function of genes specifically expressed in MSCs obtained under low density condition (Figure 5H). The results confirmed changes in cell-cycle-related genes such as M-phase and cell cycle checkpoint. When focusing on individual genes in the SAGE analysis, no significant changes were found in genes associated with senescence (Appendix A).

To evaluate the versatility of our method, we performed a verification using different lots of cells. Under the low MNC density condition, we observed a reduction in the number of hematopoietic cells remaining in the isolation process, smaller size of the isolated cells, and reduced doubling time for the different lots (Appendix A).

We analyzed surface antigens using FACS of cells at passage 2. The number of residual CD45-positive cells observed under high-density seeding conditions was greatly reduced because the number of MSCs was preferentially increased at this point. Moreover, we confirmed the expression of MSC-specific markers under both conditions (Figure 6A). Furthermore, we evaluated the differentiation potential of these cells (Figure 6B). In osteogenic differentiation, no significant differences were observed between conditions (Figure 6C). Although, in adipogenic differentiation, the percentage of FABP4-positive cells was significantly higher in the low density condition than that in the high density condition (*p* < 0.0001, Figure 6D). In chondrogenic differentiation, cells from the low density condition formed significantly larger cartilage-like tissue with a higher-density cartilage matrix inside them (*p* < 0.0001, Figure 6B,E). These results confirmed that MSCs obtained under low seeding density conditions had a higher differentiation efficiency.

## 4. Discussion

The series of studies presented in this article showed that we can improve the purity of MSCs with high proliferative ability by seeding MNCs at a comparatively low density. We further observed that colonies derived from a single cell more readily formed at low seeding densities. Additionally, as time progressed, the percentage of fast-growing cells increased because of their superior proliferative ability. Cells with superior proliferative ability were observed to increase steadily, whereas cells with inferior proliferative ability enlarged as senescence progressed. With senescence, cells become more adherent to the culture surface via development of focal adhesions and actin fibers [21,22]. Therefore, treating cells with trypsin for a shorter period ensured that the senescent cells remained in the vessel. This approach allowed the selection of MSCs with high proliferative ability to be harvested with high purity. The differentiation potential of MSCs correlates with proliferative ability [23]. Cells obtained using this method exhibited excellent quality with high proliferative ability and differentiation efficiency in subsequent cultures (Figure 7A).

In contrast, a higher seeding density of MNCs results in a limited growth area for MSCs because colonies overlap and hematopoietic cells remain. Therefore, cell density increased at an earlier stage. Our experiments suggest that high density occurs over a wide area. This results in cell-to-cell contact inhibition and produces a large number of cells with inhibited proliferation [24]. Furthermore, cells were harvested with only a small difference in proportion and with different proliferative abilities. In subsequent cultures, the overall proliferative ability was low (Figure 7B).

Some of the mechanisms described above are consistent with those described by Nakamura et al., who reported that the seeding density of synovium-derived MNCs affects the proliferative and differentiation potential of MSCs [25]. They stated that the key to maximizing the MSC yield, regardless of other tissue origins, is to form single-cell colonies, which minimize cell-to-cell contact inhibition. In addition, we proposed two insights. First, we state that adjusting the incubation time improves the purity of highly proliferative MSCs. Previous studies have shown that there are differences in the growth rate of individual MSCs [26,27]. We described the relationship in which individual MSC behavior influences overall characteristics. Second, we demonstrated a simple detachment treatment that excludes senescent cells. This treatment is known as “differential trypsinization”, which has been used for sorting cancer stem cells and epithelial cells [28,29] and is compatible with our optimized conditions. Furthermore, we believe that these methods can be applied to other cell types. Segawa et al. demonstrated that MNCs from tissues such as bone marrow, fat, synovium, and muscle formed single-cell derived colonies as MNC seeding density was reduced, showing a variety of sizes and densities [30]. In these cell types, the purity of highly proliferative MSCs is expected to increase with the progress of culture.

MSCs are a heterogeneous population showing a lineage hierarchy in terms of proliferative and differentiation potential [23,31]. When asymmetric division occurs, the hierarchy is created by dividing into cells that maintain self-renewal and those that do not [13]. These cell fates are determined by epigenetic changes [15,16]. In vivo, the accumulation of senescent cells with aging causes a senescence-associated secretory phenotype-mediated decrease in the quantity of MSC. In addition to that, factors such as BMI, diabetes, and radiation therapy can cause repetitive stress. Intracellular reactive oxygen species (ROS) accumulate in mitochondria, leading to DNA damage, arrest of the cell cycle, and dysfunction of mitochondria. These factors affect the proliferative and differentiation capacity of MSCs [32,33]. In these situations, efficient isolation of cells with high proliferative and differentiation potential is a means of increasing manufacturing stability and therapeutic efficacy. Mabuchi et al. identified LNGFR, Thy-1, and VCAM-1 as markers to isolate MSCs with high proliferative ability, termed rapidly expanding clones (RECs), which are readily isolated using cell sorters [26]. In contrast to RECs, the MSCs obtained in this study showed similarities in cell size, high proliferative ability, and high differential potential. However, gene expression pattern of REC markers was not confirmed for the MSCs obtained by us (Appendix A). Hence, we believe that the selected population differed from RECs. In the future, we will evaluate MSCs obtained by our method in vivo for clinical applications and pay particular attention to their behavior compared with that of RECs.

Comparing the cell sorter with the method used in this study, the purity of isolation was lower with our method. Thus, as mentioned above, our cells exhibit a genetic pattern different from that of RECs, and the effect of isolation tended to decrease as the number of passages increased. However, our method has the advantage of being easily implemented in production because it does not require expensive equipment and requires only a straightforward initial adjustment of the seeding density. It must also be considered that the cell sorter can adversely affect cells by shear stress. When using large numbers of MNCs, a longer processing time is required because of the processing speed, which also affects the cell growth [34,35,36].

In contrast to high proliferative MSCs, we observed that low proliferative MSCs enlarged and were SA-β-Gal positive under low MNC density condition (Figure 2A,E). Gnani et al. compared MSCs isolated from older (≥70 years) and younger (<18 years) healthy donors [33]. MSCs from older donors showed reduced colony-forming capacity, delayed cell proliferation, enlarged morphology, and increased SA-β-Gal levels. These differences increased as the culture progressed. They stated that the differences were correlated with the accumulation of DNA damage, ROS, and the activation of DNA damage response due to aging. Although the MNCs used in our experiments were derived from healthy, relatively young adults (22–34 years), MSCs with the accumulation of DNA damage and ROS in vivo could be present in a portion. Our observations under low MNC density conditions (1.25 × 10^5^ cells/cm^2^) showed that the difference in cell size from position to position was small up to 7 days, but enlargement developed after 9 days, similar to Gnani’s results. We plan to analyze the accumulation of ROS and DNA damage in these cells. Correlation with these factors will help us understand the isolation mechanisms and, thereby, predict the effectiveness of our method for unknown donors. On the other hand, under high density conditions (1.25 × 10^6^ cells/cm^2^), the distance between colonies became smaller and densified earlier. Thus, we propose that the high density condition made the senescence process invisible.

Repeated low seeding density in isolated MSCs leads to suppression of cellular senescence [37]. Compared to that observed under high seeding density (5000 cells/cm^2^), the phenotype of senescence, such as flattened morphology and p16 expression, was significantly suppressed under low seeding density conditions (50 cells/cm^2^) in the late phase of culture. ROS accumulation was especially suppressed under the low density condition. Although the relationship between seeding density and ROS accumulation remains unclear, repeated low seeding densities of isolated MSCs may lead to improved purity of highly proliferative MSCs. In a recent study regarding RECs, the Wnt pathway mediated via FDZ5 suppressed the senescence of MSCs [38]. The study also showed that FDZ5 suppressed γH2AX expression, which is presumed to suppress DNA damage and ROS accumulation. We speculate that the replacement of a large portion with these MSCs will result in a phenotype of overall suppressed senescence.

In addition to affecting the properties of MSCs as described above, MNC seeding density also affected the persistence of hematopoietic cells. We hypothesized that high seeding density conditions would enhance the interaction between monocytes or between monocytes and MSCs. Macrophage colony-stimulating factor (M-CSF) is known to be one of the key regulators in the differentiation of macrophages from monocytes [39]. It has been reported that the secretion of M-CSF by MSCs is enhanced under hypoxia, resulting in more macrophages adhering to the MSC periphery [40]. Under conditions of high seeding density of MNC, the proliferation and densification of MSCs may have led to localized oxygen depletion. This localized hypoxia may trigger enhanced secretion of M-CSF from MSCs. In our previous studies, we confirmed that the remaining CD45-positive cells were macrophages [9]. They can restrict the available growth area for MSCs, thus preventing the collection of a sufficient number of MSCs, particularly in the MSC isolation process. Furthermore, the purity of the MSCs at passage 1 in the collected cell suspensions was reduced (Appendix A). These circumstances make it difficult to seed MSCs at their most effective density in subsequent passages. Incidentally, we believe that using a low seeding density of MNCs will greatly reduce residual hematopoietic cells and stabilize manufacturing.

This study has some limitations. First, we used commercial MNCs owing to ethical considerations. The purity and proliferative potential of MNCs could vary depending on the manufacturing process and health of the donor. Therefore, it may be necessary to adjust the seeding density depending on the manufacturing procedure. Second, the basic culture conditions were based on our specific manufacturing procedures. The reagents and materials used in each manufacturing facility are different, and this was not considered.

In our study, we elucidated the mechanism by which MNC seeding density affects the quality of MSCs. Under low MNC seeding density, the proportion of highly proliferative MSCs increased over time. In contrast, low proliferative MSCs enlarged more rapidly. The senescent cells were removed using a short detachment treatment. This mechanism increases the purity of highly proliferative MSC. Based on our findings, we propose the following conditions to improve MSC quality: (1) optimizing MNC seeding density to form single-cell colonies; (2) adjusting incubation times to increase the purity of high proliferation MSC; and (3) establishing a detachment processing time that excludes senescent cells. Stabilization of proliferation will contribute to increased efficiency and stability of MSC production. We also confirmed improvement in the differentiation potential of the obtained cells. However, the correlation between seeding density and genes related to differentiation needs to be investigated. In the future, we intend to evaluate this in detail and hope that the findings will contribute to tissue engineering. Improving MSC quality is expected to be translated into improved therapeutic efficacy in cell therapy. Thus, future work would focus on verifying the clinical significance of our proposed method.

## Figures and Tables

**Figure 1 bioengineering-10-00102-f001:**
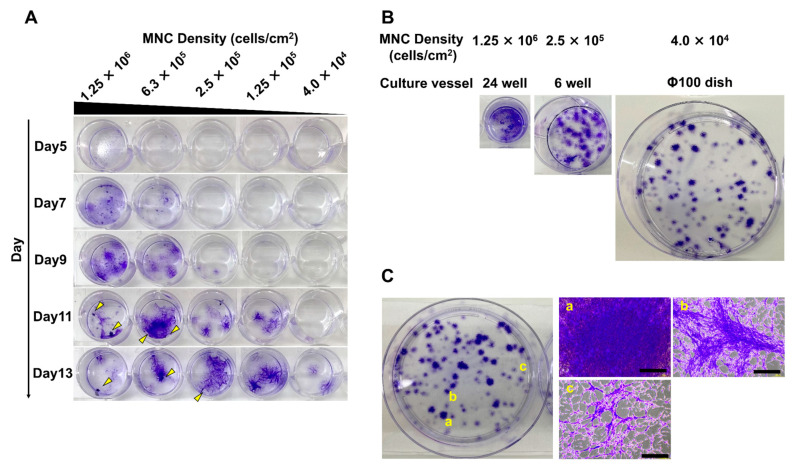
Timing of densification, distribution, and morphology of mesenchymal stem cell (MSC) colonies varied with different seeding densities of mononuclear cells (MNCs). (**A**) MNCs were cultured at various seeding densities (4.0 × 10^4^, 1.25 × 10^5^, 2.5 × 10^5^, 6.3 × 10^5^, and 1.25 × 10^6^ cells/cm^2^). MSC colonies were stained with crystal violet over time. Arrowheads indicate aggregated cells. (**B**) A fixed number of MNCs was seeded into culture vessels of different culture areas (24-well plate, 6-well plate, 100 mm dish), and cultured. Colonies were stained with crystal violet. (**C**) The pattern of colonies formed under the low MNC seeding density condition (4.0 × 10^4^ cells/cm^2^). The scale bar indicates 500 µm. (**a**) Higher density colony, (**b**) medium density colony, and (**c**) lower density colony are shown.

**Figure 2 bioengineering-10-00102-f002:**
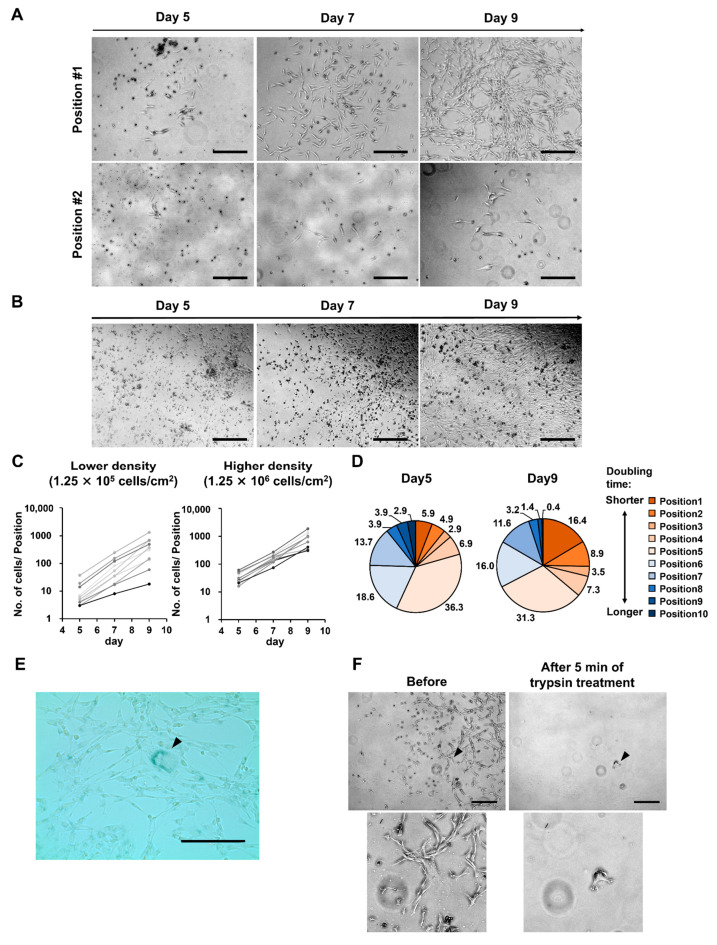
At lower mononuclear cell (MNC) seeding density, the ratio of fast-growing cells become predominant, depending on the superiority of their proliferative ability. (**A**,**B**) Each colony was observed over time using a phase-contrast microscope. The scale bar indicates 500 µm. Examples are shown for (**A**) a lower seeding density (1.25 × 10^5^ cells/cm^2^) and (**B**) a higher seeding density (1.25 × 10^6^ cells/cm^2^). (**C**) The number of cells per position of colonies formed at each seeding density is shown. (**D**) Ratio of cells in each position relative to the total cell count at a lower seeding density (1.25 × 10^5^ cells/cm^2^). Each position was color-coded based on doubling time. Numbers in the graph indicate %. (**E**) Enlarged cells seen at lower seeding density were stained with senescence-associated β-galactosidase (SA β-Gal) stain. Arrowheads point to enlarged senescent cells. The scale bar indicates 200 µm. (**F**) At lower seeding density, trypsin detachment treatment (5 min) was performed. The same position was observed microscopically before and after treatment. Arrowheads point to cells remaining after detachment treatment. The scale bar indicates 200 µm.

**Figure 3 bioengineering-10-00102-f003:**
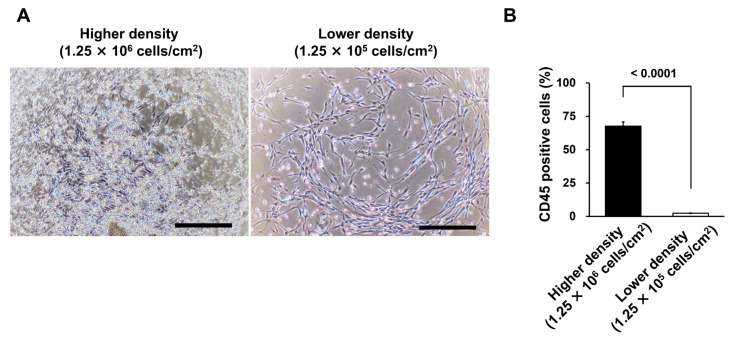
Different seeding densities of mononuclear cells (MNCs) result in different numbers of CD45-positive cells remaining. (**A**) When colonies formed, remaining hematopoietic cells were observed by phase-contrast microscopy when seeded at different seeding densities (1.25 × 10^5^ cells/cm^2^ and 1.25 × 10^6^ cells/cm^2^). The scale bar indicates 500 µm. (**B**) Data quantified as the percentage of CD45-positive cells using FACS analysis. The error bars indicate S.E. (*n* = 5). *p*-values were calculated using the Student’s *t*-test.

**Figure 4 bioengineering-10-00102-f004:**
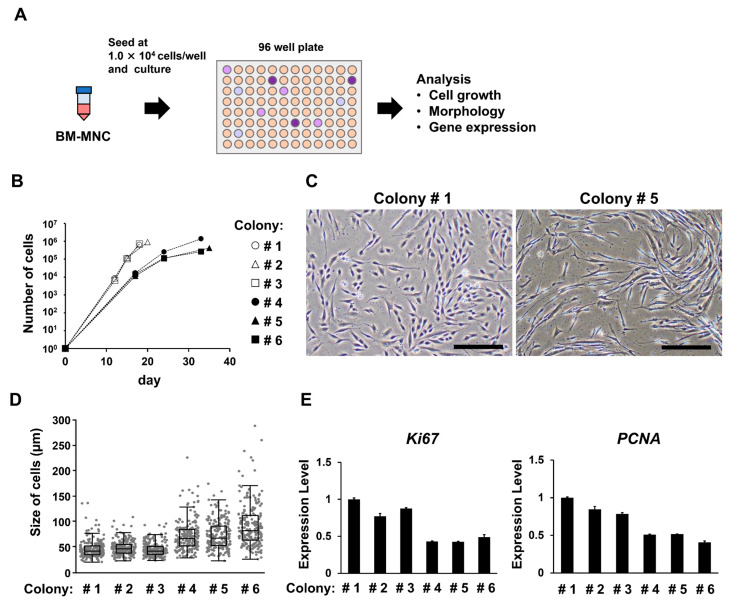
Evaluation of the relationship between the timing of colony formation and the characteristics of mesenchymal stem cells (MSCs) after passaging. (**A**) Schematic of the single cell cloning assay. Mononuclear cells (MNCs) were seeded at 1 × 10^4^ cells/well to form one colony per well, then passaged. MSCs cultured from each colony were characterized. (**B**) The number of cells harvested. (**C**) Phase-contrast microscope images of cells isolated from colonies formed earlier (colony#1) and later (colony#5). Scale bar indicates 300 µm. (**D**) Boxplot shows cell size data obtained using the microscopic images. (**E**) Data from real-time PCR analysis of *Ki67* and *PCNA* in Passage 3 cells. The error bars indicate S.E. (*n* = 3).

**Figure 5 bioengineering-10-00102-f005:**
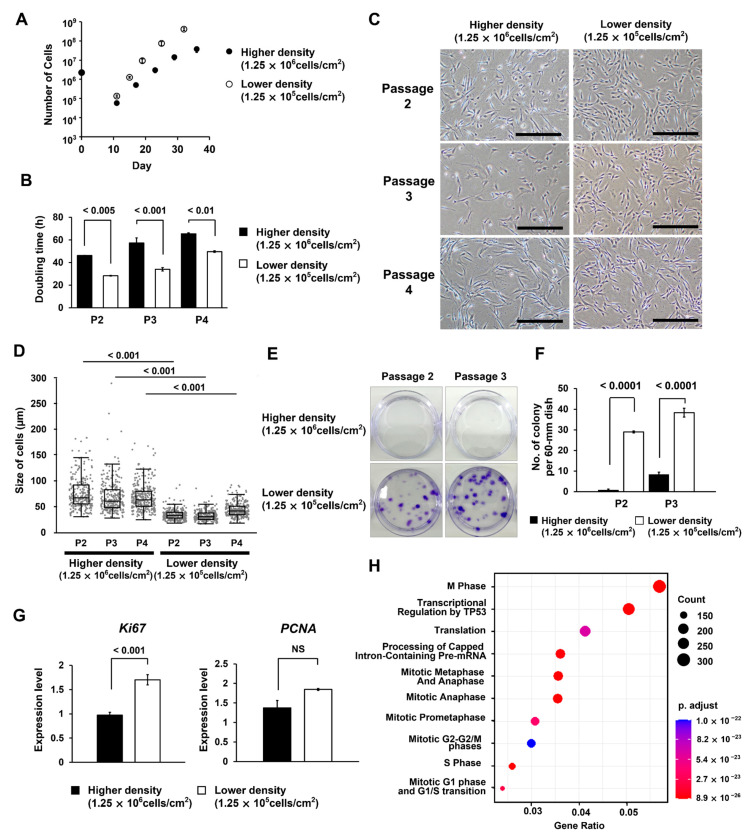
Characterization of mesenchymal stem cells (MSCs) isolated at higher (1.25 × 10^6^ cells/cm^2^) and lower (1.25 × 10^5^ cells/cm^2^) seeding densities of mononuclear cells (MNCs). (**A**) The number of cells harvested. The error bars indicate S.E. (*n* = 3). (**B**) Doubling time of cells in each passage (Passage2–4). The error bars indicate S.E. (*n* = 3). (**C**) Phase-contrast microscopy images of cells of each passage (Passage2–4). The scale bar indicates 500 µm. (**D**) Boxplot shows cell size data obtained using the microscopic images. (**E**) Colony-forming assay results. One hundred cells of passages 2 and 3 were seeded in 60 mm dishes. After being cultured for 14 d, colonies were stained with crystal violet. (**F**) Data quantifying the number of colonies formed in the colony assay. The error bars indicate S.E. (*n* = 3). (**G**) Data from real-time PCR analysis of *Ki67* and *PCNA* in Passage 2 cells. The error bars indicate S.E. (*n* = 3). *p*-values were calculated using a Student’s *t*-test. NS indicates not significant. (**H**) SAGE analysis shows the function of genes specifically expressed in MSCs obtained under low density conditions.

**Figure 6 bioengineering-10-00102-f006:**
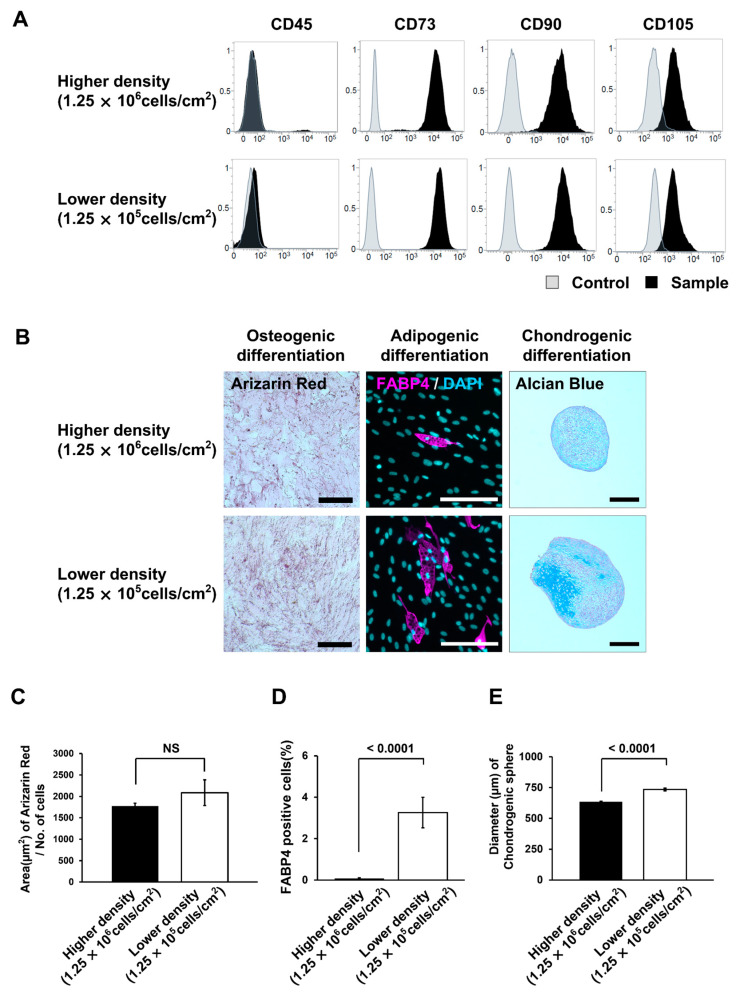
Comparison of surface antigens and differentiation potential of mesenchymal stem cells (MSCs) isolated at higher (1.25 × 10^6^ cells/cm^2^) and lower (1.25 × 10^5^ cells/cm^2^) seeding densities of mononuclear cells (MNCs). (**A**) Data for CD45, CD73, CD90, and CD105 surface antigens using fluorescence-activated cell sorting (FACS) analysis of cells (passage 2). (**B**) Images of induced differentiation of cells into osteocytes, adipocytes, and chondrocytes. The scale bar indicates 200 µm. (**C**) Alizarin-red-stained area of cells induced to differentiate into osteocytes for 21 d. The error bars indicate S.E. (*n* = 5). (**D**) Percentage of fatty-acid-binding protein (FABP)-positive cells induced to differentiate into adipocytes for 21 d. The error bars indicate S.E. (*n* = 15). (**E**) Size of spheres induced to differentiate into chondrocytes for 21 d. The error bars indicate S.E. (*n* = 6). *p*-values were calculated using a Student’s *t*-test. NS indicates not significant.

**Figure 7 bioengineering-10-00102-f007:**
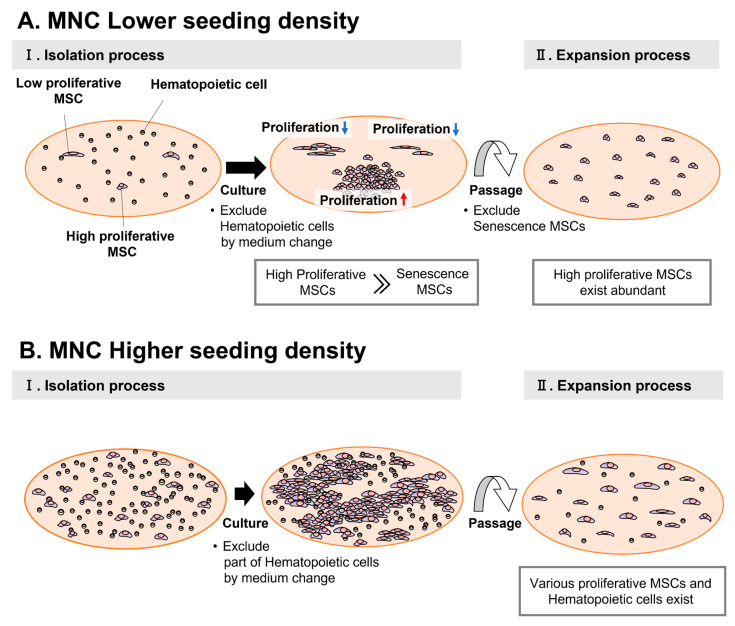
Overview of mesenchymal stem cell (MSC) colony formation at different mononuclear cell (MNC) seeding densities. (**A**) Colony-formation process under conditions of low seeding density (4.0 × 10^4^ and 1.25 × 10^5^ cells/cm^2^). (**B**) Colony-formation process under high seeding density conditions (6.3 × 10^5^ and 1.25 × 10^6^ cells/cm^2^).

**Table 1 bioengineering-10-00102-t001:** Primers used in this study.

Gene Name	Primer Sequence (5′–3′)	Product Size (bp)
*Ki67*	Forward: GGAACAGCCTCAACCATCAG	210
	Reverse: CCACTCTTTCTCCCTCCTCTC	
*PCNA*	Forward: TGGAGAACTTGGAAATGGAAA	95
	Reverse: GAACTGGTTCATTCATCTCTATGG	
*β-actin*	Forward: CGGGACCTGACTGACTACCT	96
	Reverse: CTCCTTAATGTCACGCACGA	

## Data Availability

Raw data were generated at Yamaguchi university and Shibuya corporation. Derived data supporting the findings of this study are available from the corresponding author on request.

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
