# Peer review of "Optimizing the Seeding Density of Human Mononuclear Cells to Improve the Purity of Highly Proliferative Mesenchymal Stem Cells"

_bioengineering, 2023, doi:10.3390/bioengineering10010102_

Round 1
Reviewer 1 Report (Previous Reviewer 2)
Thank you for your updates version and addressing all points in the manuscript.
Author Response
Please see the attachment.
Reviewer 2 Report (Previous Reviewer 1)
Dear,
The revised manuscript is acceptable.
Best,
Author Response
Please see the attachment.

Reviewer 3 Report (New Reviewer)
Nagai et al have studied the effect of large differences in seeding density on human BMMSC proliferation and differentiation. They show that seeding at low density increases MSC potential for single cell-derived colony formation and but also increases the number of senescent cells. Remaining hematopoietic cells were very reduced at low density. Low density allowed to select for highly proliferative cells expressing MSC markers and with higher differentiation potential. In conclusion, the study shows that very low density allows to select cells with interesting properties for therapy.
The authors use relatively low seeding density as compared to other published reports and thoroughly study the culture properties to provide an improved isolation protocol allowing homogeneous cell preparation. The experiments are well described and analyzed. However, the study would be further improved by comparing the ability of MSC proliferation and differentiation from different donors and evaluating the biological reproducibility of the results. In addition the study lacks some further analysis of the cells that are discussed (senescence, ROS). I did not find the information cited in some articles that are referred to. Finally, the manuscript sent to be evaluated was not a final version, and was with apparent modifications making it more difficult to read it
Main comments:
11/ The authors state that they used 5 lots of BMMSC (line 88). However, it is not clear how reproducible are the results obtained. Do MSC from all donors show similar behavior? What is the variability of the number of colonies, of the doubling time, of the CD45 contamination, of the number of senescent cells between the different donors?
22/ The authors should complete their visual comparison of MSC morphology by a quantitative analysis of mean FSC and means SSC values obtained by flow cytometry as in https://doi.org/10.15283/ijsc20078. In addition this article should help the authors discussing their own results in particular concerning ROS effects.
33/ How the authors explain the difference in CD45+ cells in the high density cells in Figure S1 and Figure 6A? I can see that S1 is at passage 1 and 6A at passage 2. Does this mean that at passage 2, the high density cells are free of CD45+ contaminating cells. If yes, then the improved purification capacity of low density seeding is much reduced as it only concerns passage 1. Can the authors comment on that? Hence discussion lines 456-458 is not clear when looking at figure 6A
44/ The difference in differentiation capacity between low and high density MSC should be further confirmed by RT-PCR using specific markers for each lineage
5 5/ The authors do not mention evaluation of high density MSC senescence. But they discuss it line 442-443. The authors should measure the senescence by other means than SA-bgal for example using RT-PCR for p53, p16 and p21 to be able to compare low and high density MSC.
66/ Some references are not really giving the information they are supposed to. I checked ref 8 to 10 for seeding numbers (as noted line 185), but I could only see the cell density in one reference. In addition, ref 32, cited line 410, does not refer to the effect of ROS on the proliferative capacity but on ROS effects according to the differentiation lineage.
77/ The authors discuss MSC markers LNGFR, Thy1 and VCAM (line 413). Do the low and high density MSC express them?
Author Response
Please see the attachment.

Reviewer 4 Report (New Reviewer)
This revised form of the manuscript has a much-improved English language quality.
The nature of the error bars should be included in all the relevant figures such as figures 4 and 5.
In the conclusion part:
Please correct this statement:
" Improving MSC quality is expected to be translated into improved therapeutic efficacy ".
Round 2
Reviewer 3 Report (New Reviewer)
The authors have adequately answered most of my comments. I still have one comment that does not prevent the article to be published. The answer to my comment 4 is not complete in my opinion. The difference in the differentiation potential should mostly be measured after differentiation of the two MSC populations under the same differentiating conditions. Here, with the SAGE analysis, the authors have evaluated how close the populations are to differentiated cells before the differentiation process.
This manuscript is a resubmission of an earlier submission. The following is a list of the peer review reports and author responses from that submission.
Round 1
Reviewer 1 Report
Authors investigated the relationship between seeding density and MSC colony formation, analyze the characteristics of MSCs in each colony, isolate MSCs at higher and lower seeding densities and compare their proliferative abilities and differentiation potential. It is an interesting article but there are some comments which should be considered before next step.
- Novelty was not explained in text
- Introduction is poor and author should more explain and they can use the below the reference and similar references: https://doi.org/10.1155/2022/5304860
- It would be better to add a materials section in method and materials part
-Isolated cells should be characterized by flow cytometry and ...
- Discussion is so poor and recent studies have not been discussed well in this part and there is nothing.
Reviewer 2 Report
The authors have done commendable work with a very good presentation of the research. MSC are presently a breakthrough in the treatment regime for various diseases with only a limited impediment in their ability to proliferation and differentiate appropriately in vitro and in targeted organs.
There are few points:
1. The basis of picking up concentrations for high seeding and low seeding is not very clear. An explanation about the seeding concentrations will add insight into attaining an optimal seeding concentrations.
2. Though authors have indicated the limitations of the study. It will be more suitable to derive a few general conclusion that can be applied to all MSC. Otherwise, although extensive, this study will be just a replication of the study by Nakamura et al., 2019 using a different cell line.
Thank you
Reviewer 3 Report
Optimizing the seeding density of human mononuclear cells to improve the purity of quality mesenchymal stem cells
The authors cultured human bone marrow MNCs at various seeding densities and examined the relationship between these and MSC colony formation. There are few problems need to be solved by the authors:
1. The authors are suggested to introduce some information about differentiation and senescence on MSCs.
2. The reason why the enlarged cells were SA β-Gal-positive at a low density need to be discussed in depth.
3. The mechanisms why the hematopoietic cells are reduced in low-density cells need to be discussed.
4. The authors are suggested to examine the expression of Ki67 and PCNA using western blot.
5. How did the authors determine the fast-growing cells and slow-growing cells? The authors need to introduce it in the section of “Materials and methods”.
Reviewer 4 Report
The manuscript “Optimizing the seeding density of human mononuclear cells to improve the purity of quality mesenchymal stem cells” by Nagai et al, is a research article in which the authors investigate the relationship between seeding density and colony formation of bone marrow-derived human mesenchymal stromal cells.
In the current manuscript authors claim the development of a method to increase efficiency in MSC expansion in culture.
General comment
The authors correctly pointed out that (line 51-53) “Previous reports state that MSC yield can be increased by optimizing the seeding density of mononuclear cells (MNCs) [8–10]. However, the mechanism by which this effect occurs has not been elucidated”; Line 15-16 “Optimization of the seeding density of mononuclear cells (MNCs) improves the quality of isolated MSCs. However, the underlying mechanism has not been elucidated.” Nonetheless, the current manuscript is highly descriptive and no further insight on underlying mechanism(s) is provided.
Additional issues that need to be revised:
1. Title: “the purity of quality mesenchymal stem cells”. In my opinion this part of the title is difficult comprehend.
2. Line 14: “improves the quality of isolated MSCs”. Please, provide a more accurate definition of “quality” in this context
3. Line 18: “colonies of various sizes and morphologies were formed”. The definition it is too generic.
4. Lines 26-27: “lower….longer…reduced”, comparted to?
5. Line 31: “Mesenchymal stem cells (MSCs) are stem cells”. The debate on stem cell and mesenchymal cell definition is ongoing and the terminology used from throughout several literatures to define cells isolated from different sources today is quite controversial.
6. Line 39-41: “We are currently developing a minimally invasive therapy using bone marrow-derived MSCs (BM-MSCs). This therapy is currently being evaluated in investigator-initiated clinical trials [4–6].” Define that the target disease is liver fibrosis. Not all the mentioned reference are related to “clinical trials”. “currently developing”: you should refer to recent publication.
7. Line 44: As correctly stated “the proliferative and differentiation potential of MSCs vary considerably, depending on the age and health status of the donor”. Authors address this issue also in the discussion section. However, in the current manuscript authors use MSC from a commercial source: no information on donor(s) is provided. Lot-to-lot variation should be taken in consideration. How many lots of cells did you evaluate?
8. Line 114: “Cells were cultured in differentiation medium for 21 d”. Provide more information about the medium.
9. Line 115: “FABP”. Define abbreviation upon first use
10. Line 120: “cells were seeded at 6.3×104 and 1.25×105 cells each in 15 mL tubes in a differentiation medium”. Provide information on the tubes used for cell culture.
11. Lines 153-156: “The distance between colonies increased as seeding density decreased. At a higher density of 1.25×106 cells/cm2, colonies were indistinct. At the mid-range density of 2.5×105 cells/cm2, colonies partially overlapped. At a lower density of 4.0×104 cells/cm2, individual colonies varied in size and density”. Less descriptive and more precise quantification should be provided.
12. Line 180: “Trypsin dissociation treatment tended to take longer”. Quantify.
13. Line 345: (and elsewhere in the manuscript) “MSCs isolated under the low MNC seeding density condition”. No active “isolation” procedure has been performed rather cells were simply cultured in different conditions.
14. Line 163: “we elucidated the mechanism by which the MNC seeding density affects the quality of MSCs”. Please, clearly state which is the mechanism you are referring to.
Round 2
Reviewer 1 Report
Dear Editor,
Author did not revise the manuscript properly and did not add any useful information. They did not follow my previous comments, I strongly reject this manuscript.
Reviewer 2 Report
The authors have tried to elaborate on the novelty of the work. However, this would need to be improved by supporting previous research.
Secondly, if the authors believe that their protocol can be extrapolated to other cell lines, kindly elaborate on the reason/mechanism which can substantiate the claim or may carry out a basic experiment with another cell line.
Reviewer 3 Report
In the revised article, the authors modified the manuscript and figures referred to the comments, and answered the questions comprehensively.